# Polarization and Dielectric Properties of BiFeO_3_-BaTiO_3_ Superlattice-Structured Ferroelectric Films

**DOI:** 10.3390/nano11071857

**Published:** 2021-07-19

**Authors:** Yuji Noguchi, Hiroki Matsuo

**Affiliations:** 1Division of Information and Energy, Faculty of Advanced Science and Technology, Kumamoto University, 2-39-1, Kurokami, Chuo-ku, Kumamoto 860-8555, Japan; 2International Research Organization for Advanced Science & Technology (IROAST), Kumamoto University, 2-39-1, Kurokami, Chuo-ku, Kumamoto 860-8555, Japan

**Keywords:** perovskite, ferroelectric, polarization, dielectric, BiFeO_3_, BaTiO_3_, superlattice, film, epitaxial thin film

## Abstract

Superlattice-structured epitaxial thin films composed of Mn(5%)-doped BiFeO_3_ and BaTiO_3_ with a total thickness of 600 perovskite (ABO_3_) unit cells were grown on single-crystal SrTiO_3_ substrates by pulsed laser deposition, and their polarization and dielectric properties were investigated. When the layers of Mn-BiFeO_3_ and BaTiO_3_ have over 25 ABO_3_ unit cells (*N*), the superlattice can be regarded as a simple series connection of their individual capacitors. The superlattices with an *N* of 5 or less behave as a unified ferroelectric, where the BaTiO_3_ and Mn-BiFeO_3_ layers are structurally and electronically coupled. Density functional theory calculations can explain the behavior of spontaneous polarization for the superlattices in this thin regime. We propose that a superlattice formation comprising two types of perovskite layers with different crystal symmetries opens a path to novel ferroelectrics that cannot be obtained in a solid solution system.

## 1. Introduction

Chemical tuning of the dielectric, ferroelectric, and piezoelectric properties of perovskite oxides (ABO_3_) is traditionally based on the formation of solid solutions. Lead zirconate titanate, Pb(Zr, Ti)O_3_, is representative, composed of ferroelectric PbTiO_3_ in tetragonal symmetry, and antiferroelectric PbZrO_3_ in rhombohedral symmetry [1,2]. In this system, the dielectric and piezoelectric properties are maximized near the composition-driven phase boundary [2], called the morphotropic phase boundary (MPB) [3], between the tetragonal and rhombohedral structures. The similar materials strategy has provided an extremely high piezoelectric response [4,5] in solid solutions such as Pb(Mg, Nb)O_3_–PbTiO_3_ and Pb(Zn, Nb)O_3_–PbTiO_3_, where an electric field (*E*) is considered to induce a rotation of spontaneous polarization (*P*_s_) [6].

Recently, bismuth ferrite (BiFeO_3_) [7,8] has attracted considerable attention because of its multiferroic nature [9,10], i.e., the simultaneous presence of ferroelectric *P*_s_ and an incommensurate spin cycloid structure, even at room temperature. Bulk BiFeO_3_ has a rhombohedral structure in space group *R*3*c* and possesses a large *P*_s_ along the pseudo-cubic [111]_c_ direction [11,12]. Moreover, BiFeO_3_ exhibits an extremely high Curie temperature (*T*_C_) of 830 ℃ [11,12], which can provide piezoelectric devices operating at high temperatures. In analogy to Pb(Zr, Ti)O_3_, considerable efforts have been made to investigate the solid solutions of rhombohedral BiFeO_3_ and other perovskites in tetragonal symmetry. The BiFeO_3_–BaTiO_3_ system [13,14,15,16,17] has been widely studied mainly in ceramic form because the MPB is expected to appear between rhombohedral *R*3*c* and tetragonal *P*4*mm*. Detailed structural analysis reveals that an increase in the BaTiO_3_ content causes a structural change from rhombohedral *R*3*c* to a pseudo-cubic structure [17], where the phase boundary is ambiguous at around 33% BaTiO_3_ content. Moreover, it has been reported that BiFeO_3_–BaTiO_3_ solid solutions do not have a ferroelectric nature in the BaTiO_3_ content range of 40–50% [18].

Another approach exploiting the interplay of two types of perovskite oxides is to build a superlattice by thin-film growth technology [19,20]. Epitaxially grown superlattices composed of BiFeO_3_ and BaTiO_3_ have been reported to show a high magnetoelectric coupling coefficient compared with pristine films or bulk ceramics of BiFeO_3_, where the interface plays a crucial role [21,22]. At present, ferroelectric and dielectric properties of BiFeO_3_–BaTiO_3_ superlattices have been reported in a few reports [23,24,25], and thereby the fundamental questions remain unanswered concerning how the layers of BiFeO_3_ and BaTiO_3_ are structurally and ferroelectrically coupled, and how the coupling of the two layers is activated.

In this paper, we report the crystal structure, polarization, and dielectric properties of superlattice-structured epitaxial thin films composed of BaTiO_3_ and BiFeO_3_ on single-crystal SrTiO_3_ substrates prepared by pulsed laser deposition (PLD) (Figure 1). Here, we adopted Mn(5%)-doped BiFeO_3_ instead of BiFeO_3_ to avoid a considerable influence of oxygen vacancies on the polarization and leakage current properties [23,26,27,28], because a trapping capability of oxygen vacancies by Mn^3+^ at the Fe^3+^ site, i.e., a strong attractive interaction between Mn^3+^ and oxygen vacancy, inhibits the formation of an oxygen vacancy-rich layer at the interfaces. The total number of ABO_3_ unit cells were fixed at 600, and that of the BaTiO_3_ and Mn-BiFeO_3_ layers (*N*) varied from 300 down to 1 (Figure 1b), while the average composition of the entire superlattices remained unchanged, i.e., 50% Mn-BiFeO_3_–50% BaTiO_3_. We found that the samples for an *N* greater than 25 can be regarded as a simple series connection of their individual capacitors, while those for an *N* of 5 or less behave as a unified ‘ferroelectric’, where the BaTiO_3_ and Mn-BiFeO_3_ layers are structurally and electronically coupled.

## 2. Materials and Methods

### 2.1. Experimental

Thin films of BaTiO_3_ [29] and Mn(5%)-doped BiFeO_3_ [28,30], a (Ba_0.7_Sr_0.3_)TiO_3_ buffer layer [29], and (Ba_0.1_Sr_0.9_)RuO_3_ electrodes [31] were fabricated on (100) SrTiO_3_ single-crystal substrates (5 × 5 × 1 mm^3^) by PLD (KrF excimer laser, λ = 248 nm) using ceramic targets. The details of the deposition conditions are summarized in Appendix A. Figure 1 displays the schematic of the superlattice composed of Mn-BiFeO_3_ and BaTiO_3_. The total number of ABO_3_ unit cells was fixed at 600 (Figure 1a). The number (*N*) of those in each layer forming the superlattice varied: *N* = 300, 50, 25, 10, 5, 3, and 1. Figure 1b depicts the structure of *N* = 3 as an example, where the superlattice is constructed by an alternate stacking of the thin layers of BaTiO_3_ and Mn-BiFeO_3_ with three ABO_3_ unit cells (*N* = 3). For all the samples, the Mn-BiFeO_3_ layer was deposited on the bottom electrode because of its better in-plane lattice matching with it. As a result, the layer just beneath the top electrode was the BaTiO_3_ layer. During the deposition, the following condition was adopted: a substrate temperature *T*_sub_ of 640 °C, an oxygen pressure (*P*o_2_) of 2.6 Pa, and a laser repetition rate of 1 Hz for BaTiO_3_ and 7 Hz for Mn-BiFeO_3_. The diameter of the top electrode was 0.1 mm. The polarization electric field (*P*-*E*) hysteresis properties were measured at 25 °C (3 kHz); the direction from the bottom to the top electrode was defined as positive for *E* and *P*.

High-resolution X-ray diffraction (XRD) reciprocal space maps (RSMs) were observed by using a Cu-*Kα*_1_ source. The data of the intensity profile *I_i_*(*q_x_*, *q*_z_) of reflection *i* in the reciprocal space (*q_x_*, *q*_z_) were used for the detailed analysis of the lattice parameters of the in-plane (*a*) and the out-of-plane (*c*) directions, where the parameters *a* and *c* denote those of the pseudo-cubic ABO_3_ unit cell. Throughout this paper, we adopted the pseudo-cubic notation unless otherwise stated.

### 2.2. DFT Calculations

Density functional theory (DFT) calculations were conducted using the generalized gradient approximation [32] with a plane wave basis set. We used the projector-augmented wave method [33] as implemented in the Vienna ab initio simulation package (VASP) [34]. We employed the Perdew–Burke–Ernzerhof gradient-corrected exchange correlation functional revised for solids (PBEsol) [35] and a plane wave cut-off energy of 520 eV. A Γ centered *k*-point mesh was used, and the details are provided later. Within the simplified generalized gradient approximation (GGA)+*U* approach [36], we added on-site Coulomb interaction parameters of *U**−J* of 6 eV to Fe-3*d* throughout the calculations. As the spin configuration in BiFeO_3_ can be approximated as the G-type antiferromagnet [37], we set the spin arrangement in which the adjacent Fe ions have an antiparallel spin configuration as much as possible. The experimental results for Mn-doped BiFeO_3_ films reveal that the crystal symmetry and the spontaneous polarization (*P*_s_) are not influenced by the doping of Mn up to 10%, and therefore we considered BiFeO_3_ instead of Mn-doped BiFeO_3_ for simplicity.

For building a superlattice cell, we took the following lattice constraint. Based on the experimental results of XRD for an *N* of 5 or less, the superlattice cell had a tetragonal structure with the lattice parameters of in-plane *a*_DFT_ and out-of-plane *c*_DFT_ in space group *P*4*mm*; its *a*_DFT_ was fixed at the experimental *a* of 0.3985 nm, i.e., *a*_DFT_ = *a* (experiment). The parameter *c*_DFT_ is given by the following equation, $c_{\mathrm{DFT}}$ = $Nc_{{\mathrm{BiFeO}}_{3}}+Nc_{{\mathrm{BaTiO}}_{3}}$, where $c_{{\mathrm{BiFeO}}_{3}}$ denotes the parameter *c* of the BiFeO_3_ unit cell, and $c_{{\mathrm{BaTiO}}_{3}}$ that of the BaTiO_3_ unit cell. The $c_{{\mathrm{BiFeO}}_{3}}$ and $c_{{\mathrm{BaTiO}}_{3}}$ were determined from the lattice volumes (*V*) derived from the geometrical optimizations of the BaTiO_3_ cell (5 × 5 × 5 *k*-point) and the BiFeO_3_ cell in *P*4*mm* symmetry. Considering the antiparallel spin configuration, we performed the optimization calculation of the BiFeO_3_ cell with 2$c_{{\mathrm{BiFeO}}_{3}}$ (5 × 5 × 3 *k*-point) and regarded the half cell with $c_{{\mathrm{BiFeO}}_{3}}$ as the BiFeO_3_ unit cell. For imposing the antiparallel spin configuration for *N* = 1, the long lattice with 2$c_{\mathrm{super}}$ was taken as the superlattice cell, as depicted in Figure 2a. The structural optimizations were performed under a fixed *a*_DFT_ and *c*_DFT_ with 5 × 5 × 3 *k*-point mesh for all the supercells. From the structural parameters of the optimized cell, we obtained the atomic displacements (Δ*z*) from the corresponding positions in the hypothetical non-polar paraelectric lattice. We also calculated the Born effective charges (*Z**) [38] in the superlattice cells by density-functional perturbation theory. We estimated *P*_s_, as expressed by the following equation:(1)$$P_{s}=\underset{i}{\sum }m_{i}\cdot \Delta z_{i}\cdot Z_{i}^{*}/V,$$
where $m_{i}$ denotes the site multiplicity of the constituent atom *i*, and $\Delta z_{i}\cdot Z_{i}^{*}$ is its dipole moment. The summation in Equation (1) is taken over the superlattice cell with the cell volume (*V*).

## 3. Results

### 3.1. Crystal Structure

Appendix A shows the *θ*-2*θ* XRD patterns around the 002 reflection. In addition to the peaks of the SrTiO_3_ substrate at 46.5°, the (Ba_0.1_Sr_0.9_)RuO_3_ electrodes at 46.4°, and the (Ba_0.7_Sr_0.3_)TiO_3_ buffer at 44.7°, the sample with *N* = 300 exhibits peaks individual to the layers of BaTiO_3_ and Mn-BiFeO_3_ because their layers are sufficiently thick for providing their corresponding reflections. With decreasing *N*, the integrated intensities of these peaks are weakened and eventually vanish for an *N* less than 5.

Appendix A shows the wide-area XRD-RSMs for *N* = 300 and 5. For *N* = 300 (Appendix A), the apparent reflections of 3/2 3/2 1/2 and 1/2 1/2 3/2 of Mn-BiFeO_3_ in monoclinic symmetry appear, whereas those were not observed for an *N* of 5 (Appendix A) or less. Figure 3 shows the integrated intensity of the 1/2 1/2 3/2 reflection as a function of *N*. With decreasing *N*, the intensity is weakened and then zero for *N* = 1–5. These results indicate that the monoclinic distortion, similar to the bulk (rhombohedral), is maintained in the Mn-BiFeO_3_ layer for the superlattice with *N* ≥ 10, while that is lost with *N* ≤ 5. The details of the structural analysis are described in Appendix A.

Figure 4 shows the high-resolution XRD-RSMs around the 103 reflections. For all the samples, the peak positions (*q_x_*, *q*_z_) exhibit the following features: the (Ba_0.7_Sr_0.3_)TiO_3_ buffer and the (Ba_0.1_Sr_0.9_)RuO_3_ electrodes have an apparently small *q_x_* compared with the SrTiO_3_ substrate, demonstrating that the parameter *a* of the (Ba_0.7_Sr_0.3_)TiO_3_ buffer is sufficiently expanded to the bulk value, and also that the (Ba_0.1_Sr_0.9_)RuO_3_ bottom electrode is coherently grown on the buffer. The detailed structural analysis for *N* = 300 (Figure 4a along with the 113 reflection; see Appendix A) indicates that the Mn-BiFeO_3_ layer has a rhombohedral-like monoclinic *M*_A_ structure. The splitting into two peaks of the 103 reflection of the Mn-BiFeO_3_ layer stems from the ferroelastic domain variants. With further decreasing *N*, the splitting of the Mn-BiFeO_3_ layer is smaller, and then the reflection can be regarded as a single peak for *N* = 25 and 10. At the same time, the *q*_z_ of the Mn-BFO layer with *N* = 50, 25, and 10 becomes larger than that of *N* = 300, suggesting a structural change from the *M*_A_ to monoclinic *M*_B_ phases owing to an in-plane tensile strain (see Appendix A). The experimental results, i.e., the single peak of the 103 reflection, the *q*_z_ shift, and the apparent 1/2 1/2 3/2 reflection (Figure 3), indicate that the Mn-BiFeO_3_ layer for *N* = 25 and 10 has a pseudo-tetragonal structure, with a small monoclinic (*M*_B_) distortion [39]. We note that for an *N* less than 5, the reflections from the Mn-BiFeO_3_ and BaTiO_3_ layers cannot be distinguished. These results enable us to consider that the superlattice has a unified tetragonal cell with a *c*/*a* of 1.01–1.02 as an average structure.

### 3.2. Polarization and Dielectric Properteis

Figure 5 shows the *P*-*E* loops (*E*//[001]_c_ at 3 kHz), and Figure 6a,b display the resultant remanent polarization (*P*_r_) and the maximum polarization (*P*_max_) at the highest positive *E* as a function of *N*, respectively. It is interesting to note that the superlattice samples exhibit an apparent ferroelectric polarization with an apparent *P*_r_, which is completely different from the solid solutions in the same composition (50% BaTiO_3_ content) featuring a non-ferroelectric nature [18]. The *N* = 300 sample has a *P*_r_ of 22 μC cm^−2^. The *P*-*E* loop exhibits an imprint, i.e., a shift in the negative *E* direction. This behavior is assumed to stem from a flexoelectric effect [29,40,41], where a strain gradient in the out-of-plane direction in the ferroelectric layer stabilizes the upward polarization compared with the downward one. Compared with the buffered electrode with *a* = 0.3986 nm, the BaTiO_3_ layer has the same *a*, whereas the Mn-BiFeO_3_ layer possesses a slightly small *a* = 0.3965 nm. This result indicates that a strain gradient driving the flexoelectric effect is present in the Mn-BiFeO_3_ layer adjacent to the boundary with the bottom electrode. 

From the data shown in Figure 6a–c, we think that the polarization and dielectric behavior can be divided into three regions: I. the simple series connection of the capacitors (*N* ≥ 25, see Figure 7a), II. the transition region (10 ≤ *N* < 25), and III. the unified ferroelectric regime (*N* < 10, see Figure 7b). In region I, with decreasing *N*, the hysteresis is slanted, and the resultant *P*_r_ and *P*_max_ are monotonically reduced (Figure 6a,b). We note that the relative dielectric permittivity (*ε*_r_) remains constant at ~120. This constant *ε*_r_ can be understood in terms of a simple series connection of the capacitors of the Mn-BiFeO_3_ and the BaTiO_3_ layers. Considering an *ε*_r_ of 399 for the Mn-BiFeO_3_ capacitor, and that of 93 for the BaTiO_3_ one (those were measured individually for their respective capacitors), we obtain *ε*_r_~150 (=2*ε*_r_(BaTiO_3_)·*ε*_r(_Mn-BiFeO_3)_/[*ε*_r_(BaTiO_3_)+*ε*_r_(Mn-BiFeO_3_)]). This is qualitatively in good agreement with the experiment (*ε*_r_~120). In region III, with decreasing *N*, the *P*_r_ is reduced, while the *ε*_r_ is higher.

## 4. Discussion

Figure 7 shows the schematics of the superlattice structures along with the *P*_s_ component along the out-of-plane direction (*P*_s_//[001]_c_). In region I (*N* ≥ 25), the presence of the 1/2 1/2 3/2 reflection from the Mn-BiFeO_3_ layer (Figure 3) and the polarization and dielectric properties (Figure 6) indicate that the superlattice can be regarded as the simple series connection of the capacitors of BaTiO_3_ and Mn-BiFeO_3_. In the BaTiO_3_ layer, the *P*_s_ vector is present along [001]_c_; our DFT calculations reveal that the *P*_s_ strength is 28.5 μC cm^−2^, which is close to the bulk value [42]. In contrast, the Mn-BiFeO_3_ layer has a *P*_s_ nearly along [111]_c_, and the value is reported to be 90–100 μC cm^−2^ [37]. As the polarization components along [001]_c_ in these layers are markedly different, the interface effect plays an important role. It is assumed that the interface region of several to several tens of unit cells in width needs to accommodate the difference in the direction and strength of the *P*_s_ vector across it, as in ferroelastic domain walls [43,44,45,46,47,48,49,50]. As a result, a depolarization field (*E*_dep._) is built up in the interface region, where the *E*_dep._ is present in a direction that prevents the change in the polarization component. Given that the *P*_s_ vectors are switched by an *E* application, the *P*_r_ is expected to be ~40 μC cm^−2^. The *P*_r_ of 25 μC cm^−2^ for *N* = 300 is smaller than this expected value, which is caused by a domain clamping by the *E*_dep_. In region I, the *P*_r_ is reduced when the *N* is smaller, which is because the volume fraction of the clamped domains is raised by a denser interface with the *E*_dep_.

In region III, the 1/2 1/2 3/2 reflection of the Mn-BiFeO_3_ layer is absent (Figure 3), and the polarization and dielectric properties (Figure 6) cannot be explained by the series connection of the capacitors of BaTiO_3_ and Mn-BiFeO_3_. It is reasonable to consider that the superlattice has a unified unit cell, where electronic orbitals of the BaTiO_3_ and the Mn-BiFeO_3_ layers are hybridized. In other words, these two layers are no longer distinguished, but the structural and electronic features are completely different from the solid solutions [18]. On the assumption that the superlattice has a unified unit cell (Figure 2), our DFT calculations show that the *N* = 1 cell has a *P*_s_ of 27.3 μC cm^−2^, which is close to the experimental *P*_r_ (21.6 μC cm^−2^) of *N* = 1. Moreover, the enhancement in *P*_r_ with increasing *N* (Figure 6a) can be qualitatively explained by the theoretical calculations (Figure 6d): *P*_s_ is 31.4 μC cm^−2^ for the *N* = 2 cell, and 43.7 μC cm^−2^ for the *N* = 4 cell.

Finally, we comment on an additional degree of freedom in superlattice design by adopting an unequal *N* in the BaTiO_3_ and the Mn-BiFeO_3_ layers, where material properties can be tuned by different *N*(BaTiO_3_) and *N*(Mn-BiFeO_3_). For example, we can expect that *N*(BaTiO_3_) < *N*(Mn-BiFeO_3_) delivers an enhanced *P*_s_ in a unified cell in the superlattice. Moreover, superlattice design based on different unit cell numbers is anticipated to provide a means to control the strain effect at will.

## 5. Conclusions

We investigated the crystal structure and dielectric and polarization properties of superlattice-structured epitaxial thin films composed of Mn(5%)-doped BiFeO_3_ and BaTiO_3_ with a total thickness of 600 perovskite (ABO_3_) unit cells. The number of ABO_3_ unit cell (*N*) in the layers of Mn-BiFeO_3_ and BaTiO_3_ varied from 300 down to 1. It was revealed that the superlattices for an *N* greater than 25 can be regarded as a simple series connection of their individual capacitors. In the thin regime of an *N* of five or less, the superlattice behaves as a unified ferroelectric, where the BaTiO_3_ and Mn-BiFeO_3_ layers are structurally and electronically coupled. With decreasing *N* from five to one, the *εr* is markedly enhanced, whereas the *P*_r_ is reduced. DFT calculations show that the *P*_s_ is suppressed with decreasing *N*, which is in good agreement with the experimental *P*_r_. We conclude that superlattices formed by two types of perovskite layers with different crystal symmetries represent a path to novel ferroelectrics that cannot be obtained in a solid solution system.

## Figures and Tables

**Figure 1 nanomaterials-11-01857-f001:**
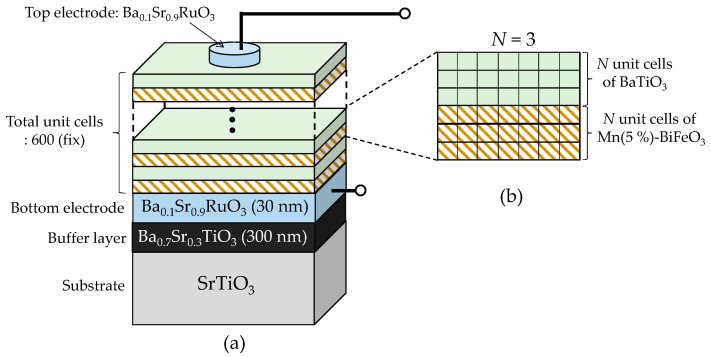
Schematics of the superlattice-structured thin film composed of layers of Mn(5%)-doped BiFeO_3_ and BaTiO_3_. The total thickness of the superlattice is fixed at 600 ABO_3_ unit cells (**a**), and the number (*N*) of ABO_3_ in the two layers varies from 300 to 1, see (**b**) for *N* = 3.

**Figure 2 nanomaterials-11-01857-f002:**
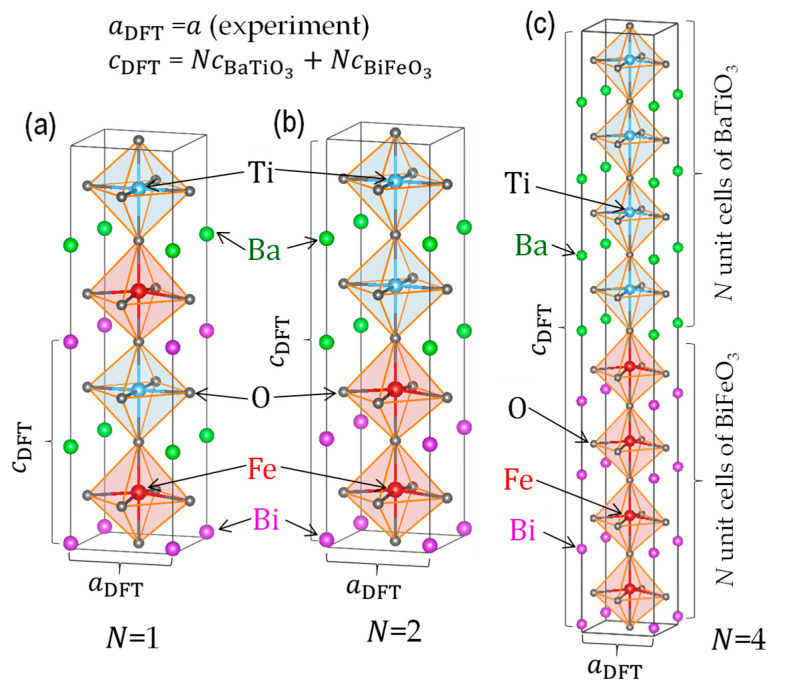
Crystal structures of the superlattice cells with (**a**) *N* = 1, (**b**) *N* = 2, and (**c**) *N* = 4 obtained from the structural optimizations by DFT calculations, where BiFeO_3_ is employed instead of Mn-BiFeO_3_ for simplicity. The in-plane lattice parameter *a*_DFT_ of the superlattice cell was fixed at the experiment: *a* (experiment) = 0.3985 nm. The out-of-plane lattice parameter *c*_DFT_ of the superlattice cell was determined from the cell volume obtained by geometrical optimizations in our preceding calculations of BiFeO_3_ and BaTiO_3_ in tetragonal *P*4*mm* symmetry.

**Figure 3 nanomaterials-11-01857-f003:**
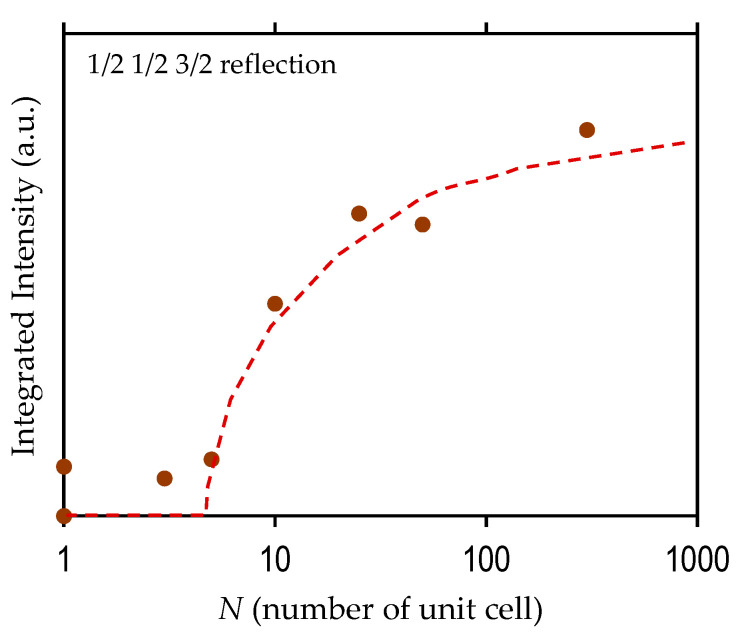
Integrated intensity of 1/2 1/2 3/2 reflection in the XRD-RSMs as a function of *N*, where *N* denotes the number of ABO_3_ unit cells in the two layers of Mn-BiFeO_3_ and BaTiO_3_ comprising the superlattice. We confirmed that the integrated intensity of 1/2 1/2 3/2 reflection of the pristine Mn-BiFeO_3_ film (300 unit cell thickness) is almost the same as that for *N* = 300.

**Figure 4 nanomaterials-11-01857-f004:**
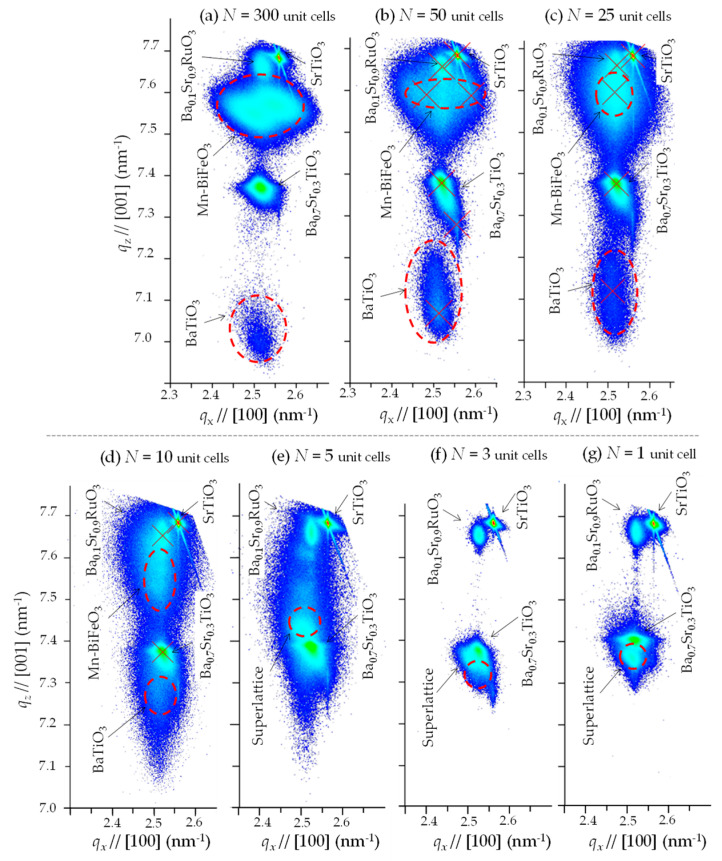
High-resolution XRD-RSMs around 103 reflection for (**a**) *N* = 300, (**b**) *N* = 50, (**c**) *N* = 25, (**d**) *N* = 10, (**e**) *N* = 5, (**f**) *N* = 3 and (**g**) *N* = 1, where the vertical axis is *q_z_*//[001] and the horizontal axis is *q_x_*//[100], where *N* denotes the number of ABO_3_ unit cells in the two layers of Mn-BiFeO_3_ and BaTiO_3_ comprising the superlattice. Here, [001] and [100] are the crystallographic directions of the (100) SrTiO_3_ substrate.

**Figure 5 nanomaterials-11-01857-f005:**
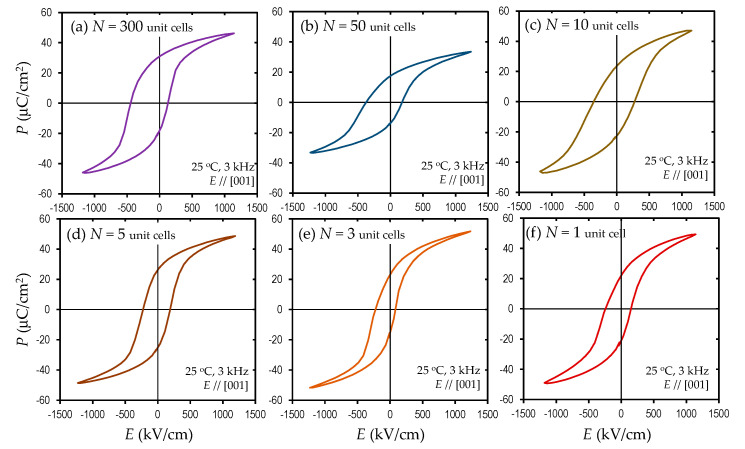
Polarization (*P*) electric field (*E*) hysteresis loops at 25 ℃, for (**a**) *N* = 300, (**b**) *N* = 50, (**c**) *N* = 10, (**d**) *N* = 5, (**e**) *N* = 3 and (**f**) *N* = 1, where an *E* of 3 kHz is applied along [001], where *N* denotes the number of ABO_3_ unit cells in the two layers of Mn-BiFeO_3_ and BaTiO_3_ comprising the superlattice.

**Figure 6 nanomaterials-11-01857-f006:**
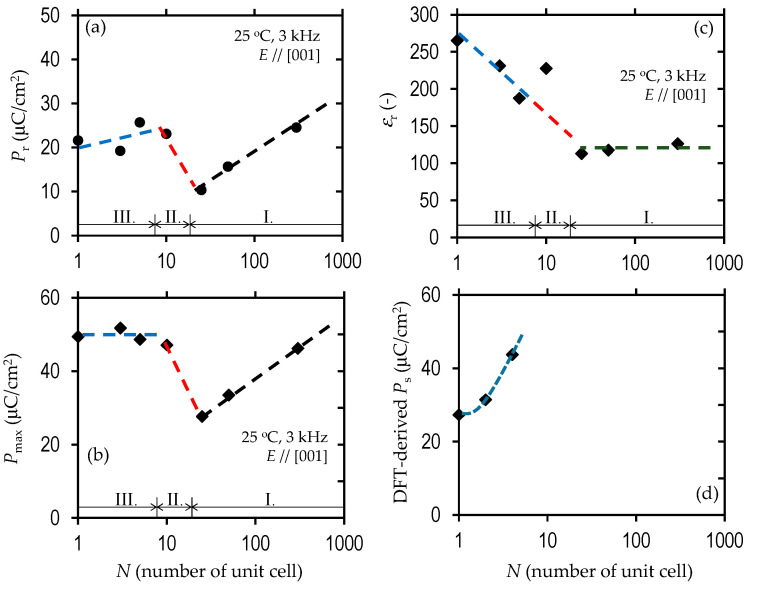
(**a**) Remanent polarization (*P*_r_), (**b**) relative dielectric permittivity (*ε*_r_), (**c**) polarization maximum at the maximum *E* (*P*_max_), and (**d**) spontaneous polarization (*P*_s_) from DFT calculations of the unified unit cells in Figure 2.

**Figure 7 nanomaterials-11-01857-f007:**
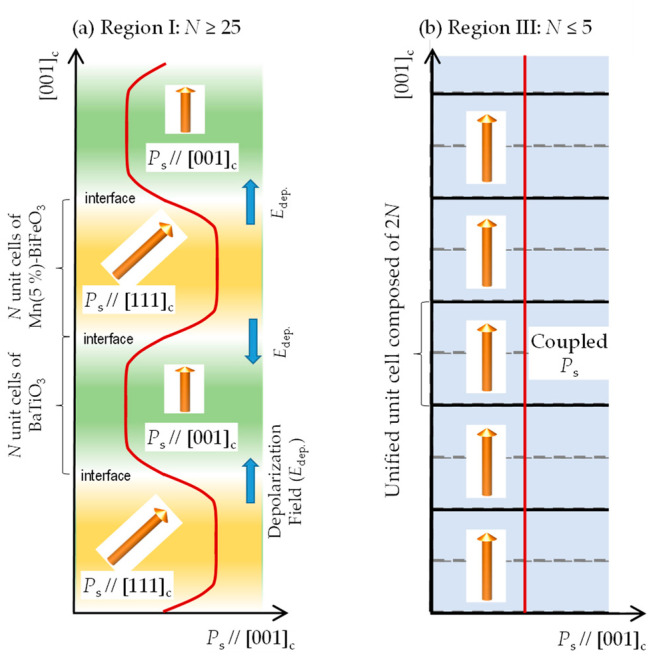
Schematics of the crystal structures of the supercells with (**a**) *N* ≥ 25, and (**b**) *N* ≤ 5, along with the [001]_c_ component of *P*_s_. In (**a**), the thickness of the layers of Mn-BiFeO_3_ and BaTiO_3_ is sufficiently thin, and thereby their polarization features are maintained inside them. In (**b**), the structural and electronic coupling of Mn-BiFeO_3_ and BaTiO_3_ is activated, and the two layers can no longer be distinguished, leading to a unified ferroelectric unit cell.

## Data Availability

The data that support the findings of this study are available upon reasonable request from the corresponding author.

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
