# Peer review of "Polarization and Dielectric Properties of BiFeO3-BaTiO3 Superlattice-Structured Ferroelectric Films"

_nanomaterials, 2021, doi:10.3390/nano11071857_

Round 1

Reviewer 1 Report

In this study, the authors report the polarization and dielectric properties of BFO/BTO superlattices. PLD was used to deposit superlattices. The authors have studied the effect of the number of ABO3 unit cells in detail. The unit cells below five and above 25 behave entirely differently. The authors conclude that superlattice formation comprising two kinds of perovskite layers with different crystal symmetries opens a path to novel ferroelectrics that cannot be obtained in a solid solution system.

The manuscript is written very well and can be accepted after addressing the following minor comments.

  1. The authors mentioned that Mn-doping was used to avoid oxygen deficiencies. Mn exhibit multiple valencies (2+. 3+, and 4+). Please elaborate on this statement as other researchers need to follow this study.
  2. Did the authors try an unequal number of unit cells for BFO and BTO layers? Could the authors comment on the effect of the unequal number of unit cells?
  3. The authors mentioned that BFO crystallizes in rhombohedral structure with monoclinic distortion for N>10 and Mb for N<10. The authors did not mention how they narrow down to Mb as its structure.? Please clarify. Could it also be MA? As 103 shows two peaks. It would be better to analyze (103) and (113) reflections together to come to the conclusion
  4. If possible, can the authors show PFM for at least two films (N = 5 and N= 300)?

Author Response

We really thank Reviewer #1 for his/her careful review together with many helpful comments. The followings are our answers along with the details of our revisions. Please see the response letter file.

Reviewer 2 Report

This manuscript demonstrates the superlattice structure and dielectric properties of Mn-doped-BiFeO3 and BaTiO3 composite lattice in which epitaxial grown thin films controlled in the number of each unit-cell period (N) from 300 to 1 and 5 ≤ N ≤ 25 region is the transition area between the activated unified coupled-ferroelectric behavior and simply mixed polarizability and ferroelectricity from both components.

More detailed descriptions of experimental conditions and evaluation for the superlattice structured thin-film composition.

  1. Is the fundamental lattice of this epitaxial BiFeO3 or BaTiO3 and the "q" value based on cubic P lattice?
  2. What represents a(super) in Figure 2?
  3. Are satellite peaks observed in Figure 4 attributed to the pure Mn-BiFeO3? Some description of the relationship between the Mn-BiFeO3 monoclinic form and the BaTiO3 tetragonal form should be required. Are the indices "212" and "211" mistaken for "221" and "221" in Figure 4?   
  4. How precisely deposit the thin films on the substrate?
  5. What is the size of the area of the thin film?
  6. Is it possible to evaluate the fluctuation of N value and uniformity of thin films?
  7. What the index for the diffraction peak assigned to "Superlattice" in Figure 3?
  8. Does the integrated intensity of [1/2 1/2 3/2] reflection reached that in pure Mn-doped-BiFeO3 increasing N value?

Author Response

We really thank Reviewer #2 for his/her careful review together with many helpful comments. The followings are our answers along with the details of our revisions. Please see the response letter file.

Reviewer 3 Report

This manuscript reports the growth of BiFeO3(Mn)/BaTiO3 superlattice thin films and their ferroelectric/dielectric properties.
Although I basically think that this is a work of high quality, I have a concern on the conclusion. Therefore, if the authors can overcome the criticisms raised below, the manuscript can be recommended for publication.

1. Formation of the BiFeO3(Mn)/BaTiO3 superlattices, as the authors claimed, was not confirmed in the manuscript. The authors should provide
an experimental evidence to prove the formation of superlattices.
In particular, for small Ns, formation of BiFeO3(Mn)-BaTiO3 solid solutions, instead of the superlattices, is concerned. 

2. For large Ns, lattice parameters, c, for the BaTiO3 films were considerably
larger than that for BaTiO3 bulk. As lattice parameters, a, for the films were
similar to that for BaTiO3 bulk, this means that the films should have larger
cell volumes as compared with that for BaTiO3 bulk. Reason for this should be provided.

3. As far as I understand, (2n+1)/2, (2n+1)/2, (2n+1)/2 reflection can also
be observed for the rhombohedral phase (R3c) in BiFeO3. So, I think that R3c
phase cannot be ruled out for the possible phase of the BiFeO3 films. 

Author Response

We really thank Reviewer #3 for his/her careful review together with many helpful comments. The followings are our answers along with the details of our revisions. Please see the response letter file.

Reviewer 4 Report

1)The first order of the theta/2theta XRD is not shown. 2)The rocking curves of the BTO and BFO which could give informations  about the crystalline quality of the samples are not shown. 3)The authors stated from XRD RSM that the qx of STO is the same as  the buffer and electrode layers(lines 173 to 176). We can see from the  figure 6 that the BSTO buffer layer and BSRO electrode are clearlly  not fully strained on STO. 4) How the exact number of the U.C and the superllatice periodicity  are determined since no clear sattillete peaks are observed on XRD and  No TEM annalysis is provided?   5)Provide details about the fabrication method and  the size of the  top electrodes. 6) To rule out the possible contribution of oxygen vacancies on the PE  loops, it is important to mesure the leakage currents of the  superlattices. 7)It seems that the used permittivities of BTO ( 93) and BFO (399) for  the series capacitors model are reversed. 8) The possible effect of  interfaces is not discussed.  

Author Response

We really thank Reviewer #4 for his/her careful review together with many helpful comments. The followings are our answers along with the details of our revisions. Please see the response letter file.

Round 2

Reviewer 3 Report

I think that the authors' response to my comments is satisfactory.

Reviewer 4 Report

The authors have satisfactorily answered all my criticisms